# Myelodysplastic Syndromes with Isolated del(5q): Value of Molecular Alterations for Diagnostic and Prognostic Assessment

**DOI:** 10.3390/cancers14225531

**Published:** 2022-11-10

**Authors:** Pamela Acha, Mar Mallo, Francesc Solé

**Affiliations:** 1MDS Group, Institut de Recerca Contra la Leucèmia Josep Carreras, ICO-Hospital Germans Trias i Pujol, Universitat Autònoma de Barcelona, 08916 Badalona, Spain; 2Microarrays Unit, Institut de Recerca Contra la Leucèmia Josep Carreras, ICO-Hospital Germans Trias i Pujol, Universitat Autònoma de Barcelona, 08916 Badalona, Spain

**Keywords:** myelodysplastic syndromes, chromosome 5q deletion, somatic mutations, cytogenetic alterations

## Abstract

**Simple Summary:**

Myelodysplastic syndromes with isolated del(5q) constitute the only MDS subtype defined by a cytogenetic alteration. The results of several clinical studies and the advances in new technologies have provided a better understanding of the biological basis of this disease. Specific genetic alterations have been found to be associated with prognosis and response to treatments. This review intends to summarize the current knowledge of the molecular background of MDS with isolated del(5q), focusing on the clinical and prognostic relevance of cytogenetic alterations and somatic mutations.

**Abstract:**

Myelodysplastic syndromes (MDS) are a group of clonal hematological neoplasms characterized by ineffective hematopoiesis in one or more bone marrow cell lineages. Consequently, patients present with variable degrees of cytopenia and dysplasia. These characteristics constitute the basis for the World Health Organization (WHO) classification criteria of MDS, among other parameters, for the current prognostic scoring system. Although nearly half of newly diagnosed patients present a cytogenetic alteration, and almost 90% of them harbor at least one somatic mutation, MDS with isolated del(5q) constitutes the only subtype clearly defined by a cytogenetic alteration. The results of several clinical studies and the advances of new technologies have allowed a better understanding of the biological basis of this disease. Therefore, since the first report of the “5q- syndrome” in 1974, changes and refinements have been made in the definition and the characteristics of the patients with MDS and del(5q). Moreover, specific genetic alterations have been found to be associated with the prognosis and response to treatments. The aim of this review is to summarize the current knowledge of the molecular background of MDS with isolated del(5q), focusing on the clinical and prognostic relevance of cytogenetic alterations and somatic mutations.

## 1. Introduction

Myelodysplastic syndromes (MDS) are a group of clonal hematological neoplasms characterized by ineffective hematopoiesis in one or more bone marrow (BM) cell lineages. Consequently, patients present with variable degrees of cytopenia and dysplasia, which are essential features for establishing a diagnosis according to the World Health Organization (WHO) classification [1,2].

Although nearly half of newly diagnosed patients present a cytogenetic alteration and almost 90% of them harbor at least one somatic mutation, MDS with isolated chromosome 5q deletion (MDS-5q) constitutes the only subtype clearly defined by a cytogenetic alteration [3,4,5,6]. The results of several clinical studies and advances in new technologies have allowed better characterization of this entity. As a consequence, since the first report of the 5q- syndrome, changes and refinements have been made in the definition and characteristics of the patients that pertain to this subtype [7]. Moreover, specific genetic alterations have been found to be associated with prognosis and response to treatments [1,2,8,9].

While preparing this review, the overview of the next WHO classification and the proposal of the International Consensus Classification (ICC) of myeloid neoplasms and acute leukemias were published [2,8]. Consequently, MDS-5q will be renamed and the inclusion criteria will be slightly modified, as will be explained in the subsequent section.

In the present review, we aimed to summarize the current knowledge of the molecular background of MDS-5q, focusing on the clinical and prognostic relevance of cytogenetic alterations and somatic mutations.

## 2. From “5q- Syndrome” to MDS-5q

In 1974, Van den Berghe et al. reported a group of three patients with refractory anemia and interstitial deletion of the long arm of chromosome 5. Such cases were later recognized as the “5q- syndrome”. Features of the syndrome included macrocytic anemia, low-normal leukocyte counts, and normal to elevated platelet counts. The BM showed erythroid hypoplasia, hypolobulated megakaryocytes, and a blast count <15% [7]. According to the French–American–British (FAB) cooperative group classification criteria, most of the patients with these characteristics pertain to the group of patients with refractory anemia [10].

It was not until the 2001 edition of the WHO classification that the 5q- syndrome was recognized as a unique and well-defined MDS subtype [11]. In addition to previously described characteristics of this syndrome, in this classification, the blast count threshold was redefined to <5%, and the absence of Auer roads was considered to define 5q- syndrome patients. In 2008, the subtype “MDS with isolated del(5q)” was introduced and the term 5q- syndrome remained restricted to a subset of cases within this category that presented with macrocytic anemia, normal or elevated platelet count, BM erythroid hypoplasia, and a blast count <5% in BM and <1% in peripheral blood (PB) [12]. In the 2017 WHO classification, these cases remained within the MDS with isolated del(5q) subtype. Additionally, the diagnosis of this subtype can be established even if there is one additional cytogenetic abnormality besides the del(5q), unless this abnormality is monosomy 7 or del(7q) [1,13]. This is based on data showing that there is no adverse effect of one chromosomal abnormality in addition to the del(5q) in such patients [14].

As mentioned previously, the overview of the next WHO classification has recently been published [2]. In this new proposal, MDS with isolated del(5q) has been renamed as “myelodysplastic neoplasm with low blast and isolated del(5q)” (abbreviated MDS-5q). The diagnostic criteria have not changed, and it is stated that although an *SF3B1* or *TP53* mutation (not multi-hit) may potentially alter the biology and/or prognosis of the disease, the presence of such mutations does not per se override the diagnosis of this entity. Regarding the ICC proposal for the classification of MDS, MDS with isolated del(5q) has been retained with no changes from the revised fourth edition of the WHO classification, although the name has been simplified to “MDS with del(5q)”. Similarly to the new WHO proposal, the ICC also specifies that *TP53* mutations are admitted in this MDS subtype unless a multi-hit state is detected [8].

It is important to remark that although del(5q) is the most frequent cytogenetic alteration in MDS and is present in roughly 20% of cytogenetic abnormal cases, only about 5% are classified in the MDS-5q category. Features such as higher blast count, alterations in chromosome 7, or a multi-hit *TP53* state impact disease prognosis and would reclassify these remaining patients into other, more aggressive, disease subtypes [2,8,15].

The changes in terms and inclusion criteria over time are produced as a consequence of the advances in technology and discoveries, which directly impact our knowledge and the management of this disease (Figure 1).

## 3. Role of Conventional Cytogenetics in MDS-5q

Conventional cytogenetics (CC) constitutes the gold standard for the genetic diagnosis and prognosis of MDS. However, fluorescence in situ hybridization (FISH) of 5q31 could be useful in cases without evidence of del(5q) by CC. When the presence of MDS-5q is suspected and/or if the cytogenetic study shows no metaphases or an aberrant karyotype with chromosome 5 is involved (no 5q deletion), it is recommended to perform FISH analysis [16,17]. Figure 2 shows the genetic studies available for the diagnosis and characterization of MDS-5q. During follow-up, genetics studies will be adapted to each patient, considering their comorbidities. A new BM aspiration, and the corresponding genetic study, will be performed on suspicion of disease evolution and no response to treatment. In the case of clonal evolution, the approach can be decided according to the general patient status.

The karyotype is a prognostic variable included in the International Prognostic Scoring System (IPSS) and the revised edition of the IPSS (IPSS-R) [18,19]. Del(5q) alone has always been considered a good prognostic variable and, in the IPSS-R, a concomitant cytogenetic alteration has been included. This change was based on a scoring system proposed by Schanz et al. based on an international data collection of 2902 patients [15]. Deletion 5q is a classical alteration detected in around 15–20% of MDS patients, with half being isolated, around 17% having an additional alteration, and 36% being part of a complex karyotype [3]. The prognostic impact of the accompanying abnormalities in del(5q) is difficult to determine because double abnormalities are highly variable. In 2011, Mallo et al. published an international collaborative study including a large series of del(5q) patients to determine the prognostic impact of adjunct prognostic abnormalities. The multivariate analysis showed that karyotype complexity was one of the main prognostic factors together with platelet count and BM blasts [14]. The good prognosis of del(5q) with one accompanying alteration was included in the MDS with the del(5q) category of the 2017 WHO classification, excluding cases harboring a chromosome 7 alteration [1].

### 3.1. Commonly Deleted Regions in Chromosome 5q

Two “commonly deleted regions” (CDR) were originally described by Boultwood and colleagues: a 1.5 Mb deletion encompassing 5q32–5q33, which was originally associated with the 5q- syndrome and better prognosis, and a more proximal CDR at 5q31. The latter was associated with other MDS subtypes and cases of acute myeloid leukemia (AML) cases, with complex karyotypes and a worse prognosis [20,21,22,23]. High-resolution techniques, such as genomic microarrays and optical genome mapping (OGM), can detect cryptic alterations accompanying the del(5q) and can help define the breakpoint. However, since high-density genomic microarrays work with DNA probes, this approach has become the most suitable technique to obtain precise del(5q) breakpoint genomic coordinates.

Most patients have large deletions that encompass both CDRs. This was corroborated by subsequent studies combining conventional cytogenetics and single nucleotide polymorphism arrays, with Mallo et al., describing a wider CDR that extended from q22.3 to q31.3. This region encompasses 14.6 Mb, while the median size of the total deletion detected in most cases is around 70 Mb [17].

Several studies have focused on the study of the 5q CDR, but in 2012, Jerez et al., published an article emphasizing the importance of the common retained region (CRR) [24]. Their work reinforces the idea that in the 5q- syndrome, the proximal and terminal regions are always retained. Thus, two CRRs were described: CRR1 for the proximal region (spanning 81.7 Mb and ending at band 5q14.2) and CRR2 for the distal region (5q34), with both being associated with disease subtypes. No CRR could be identified in other forms of MDS and AML with del(5q). As was previously described, patients with CRR had a lower number of genomic lesions and correlate with better prognosis. Additionally, this study supports the idea that genomic microarrays can add prognostic information to prognostic scoring systems as was reported by Arenillas et al., in 2013 [25]. Figure 3 shows the CDR and CRR identified for chromosome 5q.

It is widely known that additional techniques can help to describe the karyotype. Most of the complex cases carrying an apparent monosomy 5 have shown that, following studies with additional techniques such as FISH or genomic arrays, this apparent monosomy presents a partially retained 5q. As previously mentioned, genomic microarrays can help to accurately define the breakpoints [24,26].

Molecular studies revealed that haploinsufficiency of several genes (particularly *RPS14*, *CSNK1A1*, *EGR1*, *miR-14*5, and *miR*-*14a*) located in 5q CDR contribute to the pathogenesis and hematological phenotype associated with MDS-5q [27,28,29]. For example, miR-145 and miR-14a are micro RNAs that have been found to be responsible for the negative regulation of effector molecules that regulate megakaryocytic differentiation. Thus, a deficiency of these micro RNAs is responsible for the thrombosis and megakaryocytic dysplasia (hypolobulated megakaryocytes) which characterize MDS-5q [30,31].

### 3.2. Karyotyping: Present and Future Directions

Point mutations have been described as a frequent event in MDS patients. In 2011, Bejar et al. described the clinical effects of these mutations and stated that the prognosis of these patients may be driven by the association of prognostic variables. Specific genes were found to be associated with specific risk groups such as *TET2* in cases with a normal karyotype and *TP53* in cases with a complex karyotype [32]. In 2022, the IPSS-M, a prognostic scoring system based on molecular data was published. This scoring system takes into account the mutational status of 31 genes; however, it still retains the karyotype as a prognostic parameter [9].

OGM has emerged as a promising non-sequencing-based technique for high resolution genome-wide structural variant profiling. It can simplify lab workflow by reducing multiple tests. Parallel studies with standard-of-care tests have been performed in hematological neoplasms and have shown high concordance [33]. A recent study published by Yang et al. showed that OGM results changed the comprehensive cytogenetic scoring system and the IPSS-R risk groups in 21% and 17%, respectively, of their MDS patient’s cohort with an improved prediction of prognosis. Although more studies especially focused on MDS-5q are needed, the combination of OGM and next-generation sequencing (NGS) seems to be a promising approach for the evaluation of prognosis [34].

## 4. Prognostic Impact of Somatic Mutations in MDS-5q

It has been described that one-third of MDS-5q patients present with no somatic alterations, while nearly half of patients (43%) can present with an isolated mutation [35,36]. The pattern of recurrently mutated genes is similar to other MDS subtypes, except for *TP53* mutations that were found to be enriched in this subtype of patients [35,37,38]. In the subsequent section, the genes most frequently mutated in MDS-5q are described and Table 1 summarizes their biological and clinical associations and main characteristics and frequencies.

Based on data from Meggendorfer et al. [35], Malcovati et al. [39], Heuser et al. [40], and Mossner et al. [41].

### 4.1. SF3B1 Mutation

The *SF3B1* gene encodes subunit 1 of the splicing factor 3b protein complex, which is a core component of the RNA splicing machinery. Mutations in *SF3B1* have been reported in around 20% of MDS-5q cases and have been associated with a variable proportion of ring sideroblasts [5,6,39,42]. Evidence provided by several reports suggests that, in some cases of MDS-5q, the *SF3B1* mutation might precede the cytogenetic alteration [41,43,44,45]. Despite the order of acquisition of such genetic events, cases with concomitant *SF3B1* and del(5q) would still be classified within the category of MDS-5q in the WHO classification system, as well as in the ICC system [2,8].

Controversial data have been published regarding the prognosis of *SF3B1* mutations in MDS-5q patients. On one hand, a study published by Meggendorfer et al., demonstrated a significantly shorter overall survival (OS) in patients harboring both alterations compared with MDS-5q patients without *SF3B1* mutation [35]. On the other hand, no significant difference in OS was reported by Malcovati et al. when analyzing the same in their respective cohort [39].

### 4.2. DNMT3A, TET2 and ASXL1 Mutations

Mutations affecting the genes *DNMT3A*, *TET2*, and *ASXL1* genes—commonly known as DTA mutations—are frequently found in clonal hematopoiesis of indeterminate potential (CHIP), which is a non-malignant condition associated with increased risk of progression to hematologic neoplasia compared with individuals without detectable mutations [46,47]. In line with this, in mutational hierarchy studies performed by Mossner et al., DTA mutations were found to be recurrent “founder” lesions in MDS patients, including the MDS-5q cases analyzed [36,41].

*DNMT3A* codifies for DNA methyltransferase 3 alpha, which is required for genome-wide de novo methylation and is essential for the establishment of DNA methylation patterns during development [48]. On the contrary, *TET2* codifies for tet methylcytosine dioxygenase 2, which catalyzes the conversion of the modified genomic base 5-methylcytosine (5mC) into 5-hydroxymethylcytosine (5hmC) and plays a key role in active DNA demethylation. As mentioned previously, both genes are recurrently mutated in MDS. Specifically, in MDS-5q, *DNMT3A* mutations were found in roughly 18% of cases while *TET2* mutations were described in nearly 12% of patients [35].

Some studies have reported that in MDS patients, *DNMT3A* mutations were associated with a higher risk of leukemia transformation and shorter OS, but no specific study describing either phenotypic or survival associations was exclusively performed in MDS-5q patients [49,50].

A report by Scharenberg et al. described that progression in patients with low- and intermediate-1-risk del(5q) MDS is predicted by mutations in a limited number of genes, among which *TET2* is included. Specifically, 6/13 patients with evidence of disease progression presented mutations in the *TET2* gene [51]. In MDS patient cohorts including all disease subtypes, *TET2* mutations were found to be associated with shorter OS after hematopoietic stem cell transplantation and lower response rate to hypomethylating agents [9,52,53].

Located in chromosome 20q, additional sex combs 1 (*ASXL1*) codifies for a protein involved in transcriptional regulation. Mutations of mostly the frameshift type have been described in MDS patients in variable frequencies ranging from 14–24% in different cohorts [5,6,9,32,53,54]. Concretely in MDS-5q, they are less abundant and most studies describe frequencies of around 6% [35,53,54,55]. While it is a common event in early disease, Fernandez-Mercado et al. reported higher frequencies of this mutation of up to 25% among advanced cases of the disease, suggesting a role in disease progression in MDS-5q [36]. Similarly to *DNMT3A* and *TET2*, *ASXL1* mutations were mostly studied among MDS patient cohorts, including all subtypes, finding an association with worse prognosis and a shorter OS, but no specific associations were mentioned between *ASXL1* mutations and outcomes in MDS-5q cases have been described.

### 4.3. TP53 Mutations

The tumor-suppressor p53 gene (*TP53*) is located in chromosome band 17p13 and is essential for genome integrity. *TP53* encodes for the p53 protein, which is a transcription factor involved in essential cell functions, such as DNA repair, cell cycle control, apoptosis, aging, and stemness [56,57].

*TP53* gene mutations are detected in approximately 18% of MDS-5q [58,59]. It is the only mutation that was found to be significantly enriched in this MDS subtype compared with the other subtypes (18% vs. 6%) [35]. Data regarding the time of acquisition of this mutation are controversial. While it seems that there was a group of patients in which the mutation is already present in the early phases of the disease, there was another in which the *TP53* mutation arises during disease evolution, especially after treatment with lenalidomide [43,51,59].

Mutations in the *TP53* gene in MDS patients are associated with generally unfavorable outcomes, aggressive disease course, higher risk of transformation to AML, shorter overall survival (OS), and resistance to lenalidomide treatment [17,37,51,59].

Double or even triple hits in the *TP53* gene locus were already reported in 2013 by Kulasekararaj et al. [37]. A more recent study published by Bernard et al., provides new insights regarding the importance of the *TP53* allelic state (multi-hit). After studying 3324 patients, four main *TP53* mutational profiles were identified: (1) monoallelic mutations; (2) multiple mutations; (3) mutation and concomitant deletion affecting 17p; and (4) mutation and concomitant loss of heterozygosity of the 17p region. They found that two-thirds of patients with a *TP53* mutation present with multiple hits, while only one-third present with monoallelic mutations. Associations with high-risk presentation and poor outcomes were only specific to multi-hit patients, while surprisingly, monoallelic patients did not differ from *TP53* wild-type patients in outcome and response to therapy. The authors described that the *TP53* allelic state segregates patient outcomes across WHO subtypes, despite monoallelic *TP53* being enriched by MDS-5q. Moreover, they found that patients with monoallelic *TP53* mutations had longer survival compared with multi-hit patients [58].

In the 2017 edition, the WHO recommended assessing *TP53* mutational status in MDS-5q to identify high-risk cases [1]. However, the upcoming edition of the WHO classification takes into consideration new insights regarding the allelic state of this gene to redefine a specific subtype of MDS associated with the presence of multiple alterations affecting the *TP53* locus (Figure 2). This subtype is called MDS with biallelic *TP53* inactivation (MDS-bi*TP53*). However, the presence of a single *TP53* mutation (unless it is multi-hit) does not per se exclude the diagnosis of MDS-5q [2]. Similarly, the ICC proposal takes into account the *TP53* allelic state to define a new disease category called “myeloid neoplasms with mutated *TP53*”. In the case of MDS-5q, only single-hit *TP53* mutations are admitted, otherwise, the diagnosis would change to the newly mentioned category [8].

### 4.4. CSNK1A1 Mutations

Located in the CDR 5q32, *CSNK1A1* encodes for casein kinase 1A1 (CK1 α), a serine/threonine kinase that participates in many cellular processes, including growth and proliferation via the β catenin and Wnt signaling pathway, apoptosis, and response to DNA damage [60,61,62]. In 2015, a study by Kronke et al., identified CK1α as a lenalidomide target in myeloid cells and found that heterozygous deletion of *CSNK1A1* in del(5q) MDS provides a therapeutic window for selective targeting of the malignant cells [63].

Missense mutations have been reported in exons 3 and 4 in 7–10% of MDS-5q patients [35,40,44,64,65]. Detected variant allele frequency values range from 3–78% and mimic a homozygous mutation status, which is consistent with the location of the *CSNK1A1* gene and the CDR [35,40].

*CSNK1A1* mutations were found to be associated with older age and some reports show a trend towards decreased response to lenalidomide, but no independent prognostic impact on OS has been described to date [40,60]. In a study performed by Meggendorfer et al., *CSNK1A1* mutations were found to co-occur with *SF3B1* mutations in 42% of the cases [35].

### 4.5. JAK2 Mutations

Janus kinase 2 (*JAK2*) encodes a non-receptor tyrosine kinase that plays a central role in cytokine and growth factor signaling. Somatic mutations in *JAK2* constitute a major diagnostic criterion for myeloproliferative neoplasms (MPN) and are found in approximately 95% of polycythemia vera cases and 50% of essential thrombocythemia and primary myelofibrosis [2,9,66,67].

Mutations in this gene, specifically the V617F hotspot, were reported in approximately 6% of patients with MDS-5q and were found to correlate with higher platelet counts when compared with *JAK2* wild-type patients [35,55,68]. Sangiorgio et al., performed a detailed microscopic analysis of BM aspirates of MDS-5q cases with concomitant *JAK2* mutations and found greater reticulin fibrosis in mutated cases. Additionally, they found a combination of hypolobulated megakaryocytes (typically found in MDS-5q) and large forms with hyperlobulated nuclei, which are commonly seen in MPN [69].

Although the phenotypic characteristics have been described, no significant differences in OS or disease progression were found in such MDS-5q *JAK2* mutated cases when compared with *JAK2* wild-type cases [55,69].

## 5. Clonal Evolution

As mentioned previously, several authors have described the commonly mutated genes and concomitant copy number alterations in MDS-5q, but few studies have explored clonal evolution in this specific subtype of MDS.

The first systematic study providing molecular monitoring of long-term serial follow-up samples in a significant cohort of patients was by Mossner et al. [41]. As in most of the subsequent publications, such clonal evolution studies are based on bulk sequencing (exome or gene panel), in which clonal composition and evolutionary patterns are reconstructed based on variant allele frequency values of the detected mutations. The authors described that MDS “founder” lesions recurrently affected genes involved in the regulation of DNA methylation (e.g., *TET2*, *DNMT3A*), chromatin remodeling (e.g., *ASXL1*), or RNA splicing (e.g., *SF3B1*), and that del(5q) was acquired as a secondary lesion or constituted a minor independent clone in 62% of patients classified as MDS-5q. This is in contrast to previous studies proposing del(5q) as the initiating lesion in such patients [44]. In line with this, single-cell studies performed by our group demonstrated that in some MDS-5q cases, del(5q) can appear as the initiating lesion, while it can appear as a secondary hit in other cases [43].

As expected, the emergence and disappearance of specific clones in the BM are frequently correlated with changes in the clinical features in PB, such as hemoglobin and platelet levels. Moreover, it has been described that, in almost all cases, treatment with lenalidomide induced an effective reduction of cells carrying del(5q), however, it did not induce complete molecular remission of all clones carrying typical MDS mutations [41]. Furthermore, loss of response to lenalidomide is correlated with the gradual growth of a non-related clonal population already detectable at low levels before treatment or the expansion of a descendent from the original clone of the diagnosis [70].

In another longitudinal study, Scharenberg et al. described that 37% of their MDS-5q cohort progressed to either higher-risk MDS or transformed into AML in a median of 85 months after diagnosis. Interestingly, they found that all the cases harbored recurrent mutations in *TP53*, *TET2,* or *RUNX1* in addition to del(5q) [51]. Thus, several patients showed an increased allele burden and gains of new mutations during the course of the disease and treatment. Particularly, the acquisition of *TP53* mutations was relatively common in the progression of patients treated with lenalidomide, with some of them exhibiting more than one *TP53* mutation.

In general, all the above-mentioned studies agreed that both linear and branched evolutionary patterns occur with and without disease-modifying treatments, and subclones that acquire additional mutations associated with treatment resistance or disease progression can be detected months before clinical changes become apparent [41,43,51,70].

## 6. Conclusions

Based on our understanding of MDS-5q, together with the changes in the inclusion criteria, the evaluation of prognosis evaluation and the management of the disease are clearly in line with the progress of molecular genetics, which at the same time are linked to the advances in technology and scientific discoveries.

With the arrival of the IPSS-M and the newly proposed classifications for MDS, NGS techniques are mandatory for correct disease classification and assessment of prognosis. However, the approaches to financing health care are extremely diverse and are country-specific, and therefore, there may still be situations in which NGS remains restricted to potentially guiding therapeutic decisions, such as treatment intensity or hematopoietic stem cell transplantation.

Although many advances have been achieved, especially in the last decade, unanswered questions remain. Techniques such as OGM and new single-cell techniques together with new clinical trials are just some future steps to better understanding this disease and ultimately improving patient care.

## Figures and Tables

**Figure 1 cancers-14-05531-f001:**
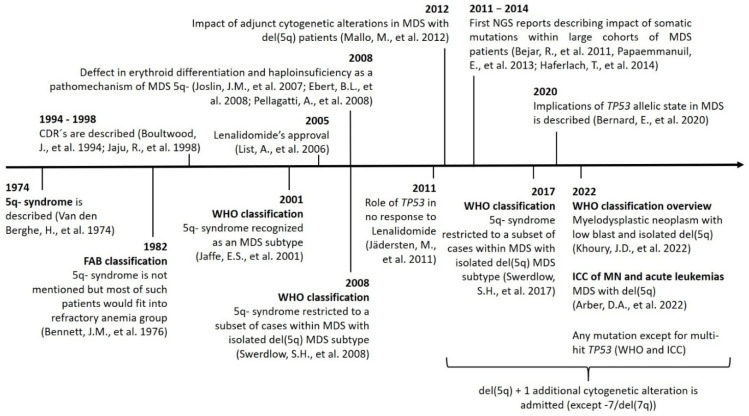
Timeline showing the main discoveries involving MDS-5q and the changes in nomenclature and inclusion criteria. Abbreviations: CDR, commonly deleted region; chr, chromosome; ICC: International Consensus Classification; MN, myeloid neoplasms; NGS, next-generation sequencing.

**Figure 2 cancers-14-05531-f002:**
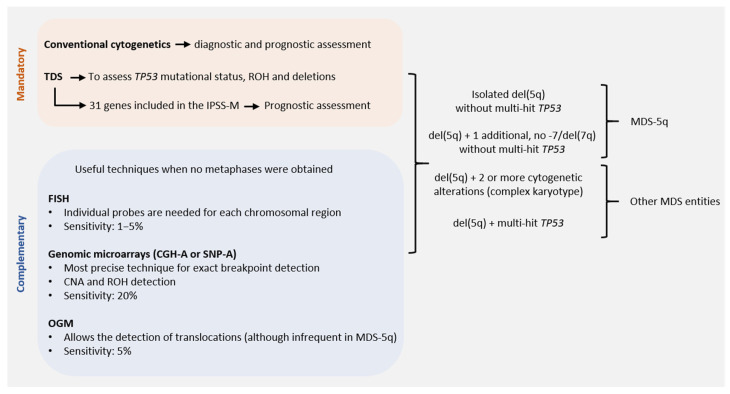
Genetic studies of MDS-5q according to the new diagnostic and prognostic guidelines: techniques available for correct diagnostic and prognostic assessment of MDS-5q according to the criteria of the next World Health Organization (WHO) classification, the proposal of the International Consensus Classification (ICC) and the Molecular International Prognosis Scoring System (IPSS-M). Abbreviations: CGH-A, comparative genomic hybridization; CNA, copy number alteration; FISH, fluorescence in situ hybridization; OGM, optical genome mapping; ROH, region of homozygosity; SNP-A, single nucleotide polymorphism array; TDS, targeted gene sequencing (assuming the use of probes that allow the detection of small CNA and ROH. Otherwise, SNP-A would be recommended to assess CNA and ROH in *TP53* for accurate diagnostic and prognostic assessment).

**Figure 3 cancers-14-05531-f003:**
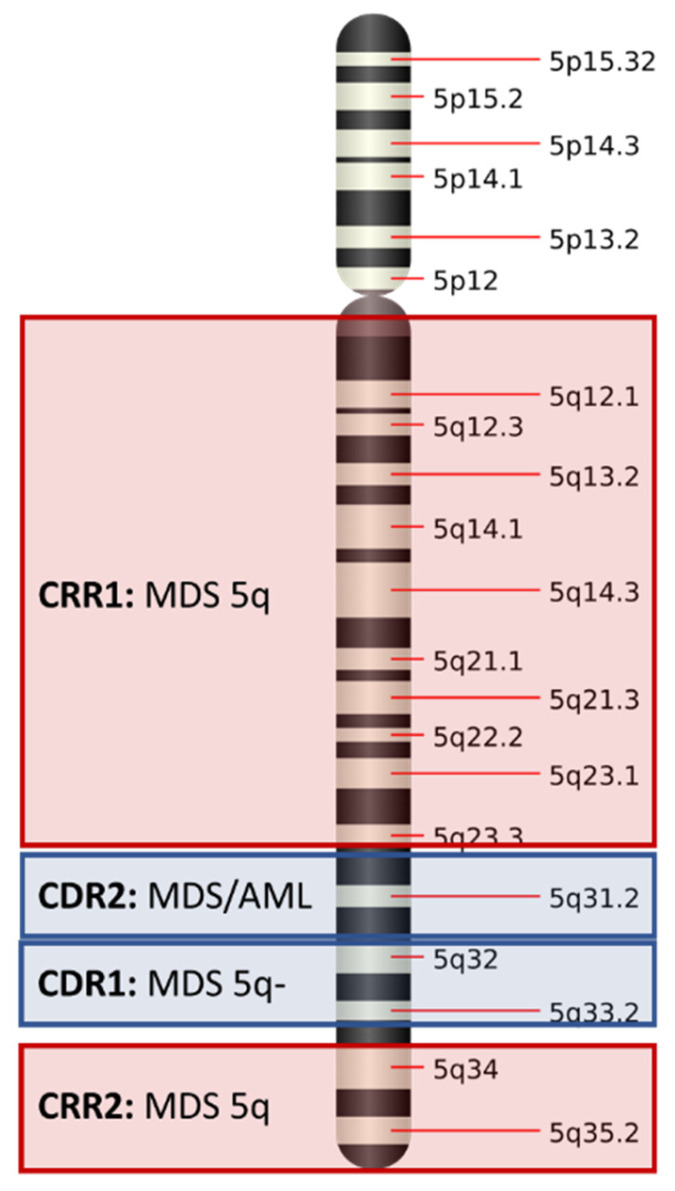
Commonly deleted regions (CDR) and commonly retained regions (CRR) in chromosome 5q.

**Table 1 cancers-14-05531-t001:** Recurrently mutated genes in MDS-5q: clinical and biological correlations.

Gene	Pathway/Function	Frequency	Clinical and Biological Correlations
*SF3B1*	Splicing factor	19–20%	Associated with RS Controvert data regarding outcome of concomitant *SF3B1* mutation and del(5q)
*DNMT3A*	DNA methylation	18%	Recurrent founder lesion (DTA mutations)
*TP53*	Checkpoint/cell cycle	18%	Aggressive disease course Higher risk of transformation to AML Shorter OS Resistance to lenalidomide treatment
*TET2*	DNA methylation	12%	Recurrent founder lesion (DTA mutations)
*CSNK1A1*	Proliferation, apoptosis, DNA damage response	7–10%	Associated with older age
*ASXL1*	Chromatin modification	6%	Recurrent founder lesion (DTA mutations)
*JAK2*	Tyrosine kinase	6%	Associated with elevated platelet counts

Abbreviations: AML, acute myeloid leukemia; DTA, *DNMT3A*, *TET2*, and *ASXL1*; OS, overall survival; RS, ring sideroblasts.

## Data Availability

Not applicable.

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
