# Peer review of "Myelodysplastic Syndromes with Isolated del(5q): Value of Molecular Alterations for Diagnostic and Prognostic Assessment"

_cancers, 2022, doi:10.3390/cancers14225531_

Round 1

Reviewer 1 Report

the authors present an updated review of MDS with isolated 5q deletion.

It references the newest WHO and ICC classifications and reviews additional genetic abnormalities.

It is comprehensive and a useful resource.

It appears not to be written by a native English speaker and could benefit from minor edits. 

Author Response

Reviewer 1 comments:

The authors present an updated review of MDS with isolated 5q deletion.

It references the newest WHO and ICC classifications and reviews additional genetic abnormalities.

It is comprehensive and a useful resource.

Point 1: It appears not to be written by a native English speaker and could benefit from minor edits

Response 1: The manuscript has been reviewed by a native speaker and has been corrected accordingly.

Reviewer 2 Report

This is a well written review on MDS with isolated del5q with a focus on additional molecular alterations already also adressing the newly published classifications and prognostic tools.

Some points for possible amendments:

Although "clinical relevance" is mentioned as topic in the abstract this should  be further elaborated: how often during the course of the disease should genetics be monitored ? should all methods be applied in all circumstances? which consequences should be taken upon detection of clonal evolution ?

some suggestions for Figure 1: 

in upper part add a mention of elucidation of pathomechansims (ref 27-29)

when mentioning allelic state of tp53 - also include initial data on tp53

only selected items have references: need to give references to other points as well,  to have a clearer picture probably only ref number of ref list

Minor points:

Line 39: ref 3 is ICC not WHO

Line 196 Prognostic impact

Author Response

Reviewer 2 comments:

This is a well written review on MDS with isolated del5q with a focus on additional molecular alterations already also adressing the newly published classifications and prognostic tools.

Some points for possible amendments:

Point 1: Although "clinical relevance" is mentioned as topic in the abstract this should be further elaborated: how often during the course of the disease should genetics be monitored? should all methods be applied in all circumstances? which consequences should be taken upon detection of clonal evolution ?

Response 1: New information regarding follow-up approach has been added to the manuscript (lines 215-219).

Point 2: Some suggestions for Figure 1:

In upper part add a mention of elucidation of pathomechansims (ref 27-29)

When mentioning allelic state of tp53 - also include initial data on tp53

Only selected items have references: need to give references to other points as well,  to have a clearer picture probably only ref number of ref list

Response 2: Related to Figure 1:

  • A timeline point about pathomechanisms has been added.
  • Regarding TP53, Jädersten et al study, which revealed the poor response of MDS 5q patients with TP53 mutation to lenalidomide treatment, has been added.
  • Lastly, reference numbers have been added.

Point 3: Line 39: ref 3 is ICC not WHO

Response 3: References have been redistributed.

Point 4: Line 196 Prognostic impact

Response 4: The header from (old) line 196 has been changed.